# Community diversity and habitat structure shape the repertoire of extracellular proteins in bacteria

Marc Garcia-Garcera [1,2]* & Eduardo P.C. Rocha [1]*

We test the hypothesis that the frequency and cost of extracellular proteins produced by bacteria, which often depend on cooperative processes, vary with habitat structure and community diversity. The integration of the environmental distribution of bacteria (using 16S datasets) and their genomes shows that bacteria living in more structured habitats encode more extracellular proteins. In contrast, the effect of community diversity depends on protein function: it's positive for proteins implicated in antagonistic interactions and negative for those involved in nutrient acquisition. Extracellular proteins are costly and endure stronger selective pressure for low cost and for low diffusivity in less structured habitats and in more diverse communities. Finally, Bacteria found in multiple types of habitats, including host-associated generalists, encode more extracellular proteins than niche-restricted bacteria. These results show that ecological variables, notably habitat structure and community diversity, shape the evolution of the repertoires of genes encoding extracellular proteins and thus affect the ability of bacteria to manipulate their environment.

---

[1] Microbial Evolutionary Genomics, Institut Pasteur, CNRS, UMR3525, 28, rue Dr Roux, 75015 Paris, France. [2]Present address: Department of Fundamental Microbiology, University of Lausanne, Batiment Biophore, Quartier SORGE, 1003 Lausanne, Switzerland. *email: marc.garciagarcera@unil.ch; erocha@pasteur.fr

Bacteria use secretion of proteins for numerous biotic and abiotic interactions[1]. Secreted enzymes can be used to scavenge for nutrients. Secreted toxins are involved in defense against protozoa, virulence towards humans, and antagonistic interactions with other bacteria[2–4]. Other small metabolites are also implicated in interactions among bacteria and of bacteria with eukaryotes[5], but they will not be analysed here. The functions provided by extracellular proteins come at a significant cost. Extracellular proteins are costly to secrete, a process usually requiring energy and specific complex machineries at the cell envelope[6,7]. Once outside of the cell, these proteins are lost to the producing individual and their amino acids cannot easily be recycled. Furthermore, extracellular proteins can act as public goods that benefit other bacteria in the community, those with the ability to benefit from its function, even when they have not contributed for their production[8]. For example, enzymes involved in the degradation of fatty acids or complex polysaccharides produce simple compounds that can be used by numerous bacteria[9]. Similarly, toxins protecting bacteria from grazing protozoa provide a public service to bacteria that may have incurred no cost in their production[10]. Processes where individuals produce public goods are prone to social exploitation because bacteria that do not pay the cost of producing the extracellular protein but reap the associated benefit may be at an advantage relative to the producer that pays the cost[11–13]. As a result, cheaters may rise in frequency in the population and lead to the collapse of the cooperative process underlying the production of the public good. This is analogous to the classical tragedy of the commons in economic theory, and has received extensive attention in evolutionary biology[14,15].

Bacteria produce a large number of extracellular proteins. A recent study predicted that 6% of the gene families of the pan-genomes of Proteobacteria encode extracellular proteins[16]. The genes encoding these proteins have some peculiarities. First, they tend to be gained and lost at a high rate, relative to other genes, and are often encoded in mobile genetic elements (MGEs), most notably in plasmids[13,17]. The acquisition of genes encoding extracellular proteins stabilizes cooperative processes because bacteria lacking these genes—potential cheaters—may become co-operators upon infection with the MGE[18,19]. On the other hand, extracellular proteins are involved in specific processes—like competition and scavenging—that are under complex types of natural selection, including frequency-dependent, fluctuating, and intermittent selection[20–22]. Second, extracellular proteins tend to be composed of amino acids that are less expensive than average to synthetize[13,16]. This fits the intuition that the low cost of these proteins should be under strong selection because they cannot be recycled and because cooperation is more likely to evolve when the cost of the process—the extracellular protein—is low relative to the benefit acquired by the kin in the community (Hamilton's rule[23]). These previous results suggest that the cost of production and the benefit that these proteins can bring back to the producer are very important determinants of the selection for the maintenance (and expression) of the genes encoding the extracellular proteins.

Selection for the production of public goods depends on habitat structure and community diversity. Habitat (spatial) structure tends to stabilize cooperation processes because highly viscous, or fragmented, habitats increase the probability that neighbors of the producing bacterium are kin and also producers of the same proteins[24]. The patches of genetically related bacteria created by these structured habitats also result in local environments that have low functional diversity[25], and bacteria may need to produce more extracellular proteins as a compensation mechanism. The frequency of genes encoding extracellular proteins is thus expected to increase with habitat structure.

How the frequency of genes encoding extracellular proteins varies in function of community diversity is harder to predict. Highly diverse communities are more likely to include individuals capable of exploiting cooperative processes, e.g., profit from the action of degradative enzymes produced by other Bacteria. They are also more likely to have high functional diversity[26], i.e., require a lower input of functions per species. This is the basis of the Black Queen model, where bacteria may evolve to lose genes in order to produce complementary functions in a community[27]. Both effects might lower selection for large repertoires of genes encoding extracellular proteins. On the other hand, diverse communities are also more likely to include antagonistic interactions between bacteria, which often implicate the use of secreted proteins such as bacteriocins[22].

Many theoretical studies have addressed the conditions for the stabilization of cooperation processes by production of public goods (see West et al.[14]). A few works have experimentally tested these ideas with secreted proteins or siderophores[19]. But there has been little work on how these conditions affect the repertoire of extracellular proteins of natural microbial populations (for a recent work on siderophores, see Butaitė et al.[28]). To assess the role of environmental traits in the distribution of genes encoding extracellular proteins, we identified their genes in genomes of Bacteria. We focused on genomic data, because detection of extracellular proteins is accurate in genomes[29], but not in meta-genomes. We then searched for these bacteria in 16S rRNA environmental datasets. We adapted a previously published method to assess habitat structure[30], and used the effective number of species to characterize community diversity. We then integrated this information to search for associations between the frequency of genes encoding extracellular proteins in genomes, the structure of habitats, and the diversity of communities.

Our analysis shows that bacteria living in more structured environments encode for more extracellular proteins and higher diffusibility. However, the association of the latter traits with habitat diversity is more complex, since it depends on protein function. Finally, our observations suggest that bacteria able to colonize multiple environments encode a greater and more diverse repertoire of extracellular proteins. Overall, our findings indicate that the features of habitat structure and community diversity shape the ability of bacteria to interact with their environment.

## Results

**Distribution of genes encoding extracellular proteins**. We identified 109,671 genes encoding extracellular proteins—the secretome—in 8294 out of 10,673 replicons from 5397 bacterial genomes. We did not include in this analysis outer membrane proteins, because they are closely associated with the producer cell, even if our previous works suggest they share many traits with extracellular proteins[16]. Extracellular proteins correspond to 1.58% of the proteins for which we could predict a cellular localization (Supplementary Fig. S1). This is very similar to the frequency of such proteins previously identified in Proteobacteria (1.8%, Nogueira et al.[16]). One should note that because these genes are usually at low frequency in the species genomes, they tend to account for a much larger fraction of the pan-genome (e.g., 5.3% in *Mycobacterium haemophilus*). Their number is strongly correlated with genome size (Spearman rho ($\rho$): 0.81, $P < 10^{-16}$). To highlight the functions of these proteins, we searched for sequence similarity between each extracellular protein and the database eggNOG v. 4.5 (Fig. 1). This shows that these proteins have very diverse functions.

Around 40% (41,373) of the extracellular proteins lacked significant hits and could not be annotated this way. Their

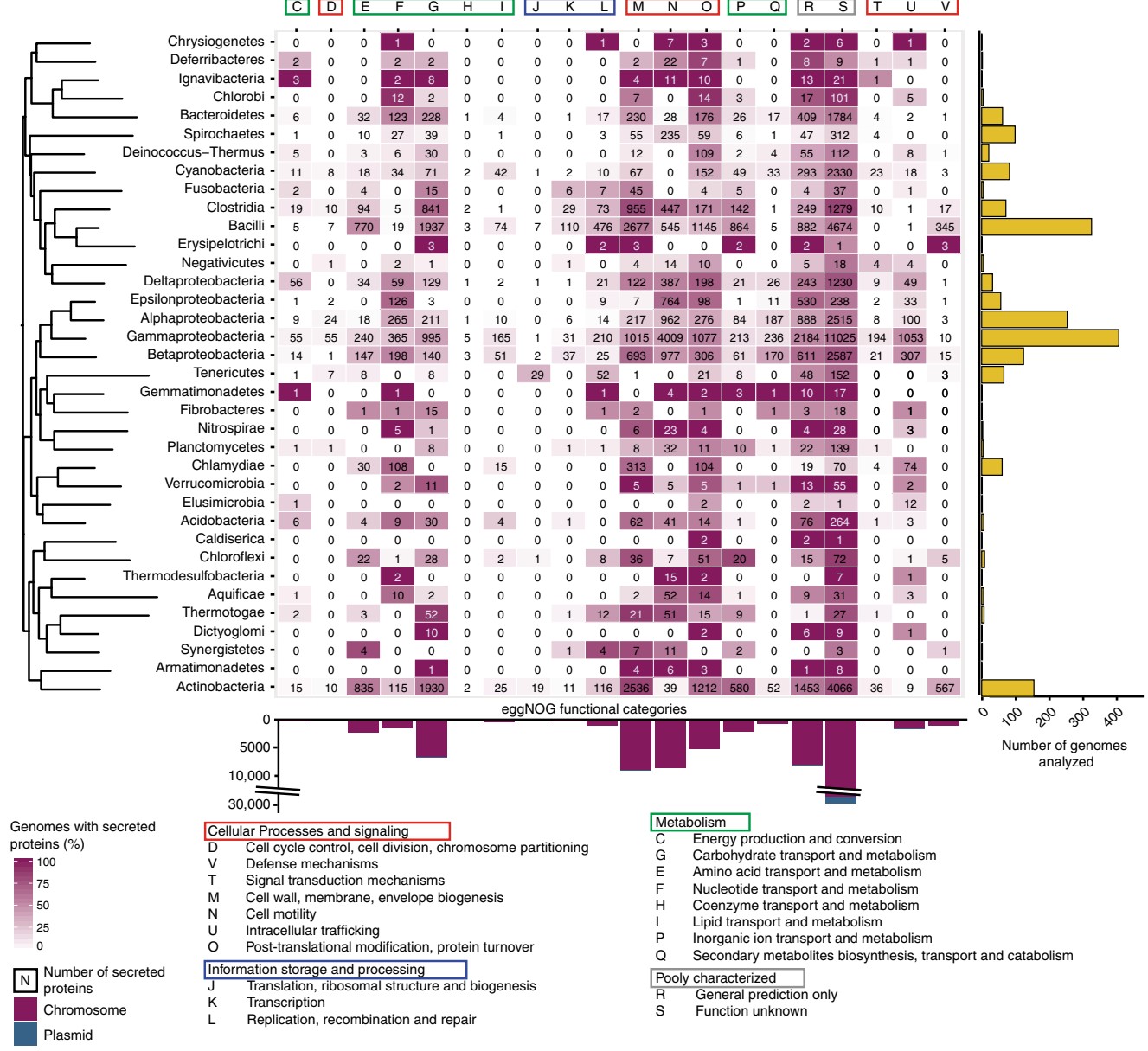

**Fig. 1 Number of genes encoding extracellular proteins per large taxonomic group and in terms of COG protein function.** The colors of the heatmap indicate the percentage of genomes in the clade encoding extracellular proteins associated to specific functional groups. In the absence of a consensual phylogenetic tree of prokaryotes, we constructed a phylogenetic tree using the 16S rRNA phylogeny. The bar plots resume the number of genomes analysed (right) and of genes encoding extracellular proteins (bottom). Supplementary Fig. 3 shows the dotplot representing the relative frequencies of each COG category in chromosomes and plasmids.

function remained unknown. As expected, proteins involved in replication, transcription and cell division were significantly under-represented among extracellular proteins (one-way Wilcoxon rank test, $P < 10^{-48}$). In contrast, three categories related with metabolic activities—on carbohydrates, amino acids and nucleic acids—were over-represented (Kruskal-Wallis test, $P < 10^{-18}$). Other over-represented functional categories in the secretome include post-transcriptional modification and protein turnover, cell mobility and secretion, and biogenesis of the cell envelope ($P < 10^{-12}$, same test). Finally, the frequency of extracellular proteins was higher in plasmids (36.3 Mb) than in chromosomes (11.8 Mb, Wilcoxon rank test, $P = 1.29 \times 10^{-89}$, Supplementary Fig. 2), as previously reported[13,16]. This tendency is valid across all categories of functions of extracellular proteins for which enough data is available (Supplementary Fig. 3). One

should point out that since chromosomes account for the majority of genes in genomes, they actually contain more genes encoding extracellular proteins, albeit at much smaller densities than plasmids. In short, our dataset reproduces previous results showing that extracellular proteins are abundant, functionally diverse, and are often encoded in mobile elements, which should favor their horizontal transfer.

**High habitat structure favors extracellular proteins.** Highly structured habitats tend to favor cooperation, leading to the hypothesis that Bacteria present in such habitats encode more extracellular proteins. To test it, we analysed around ten thousand 16S rRNA datasets from five independent broad categories of habitats—water, sediment, sludge, soil and host-associated—inspired by a previous work on small extracellular compounds

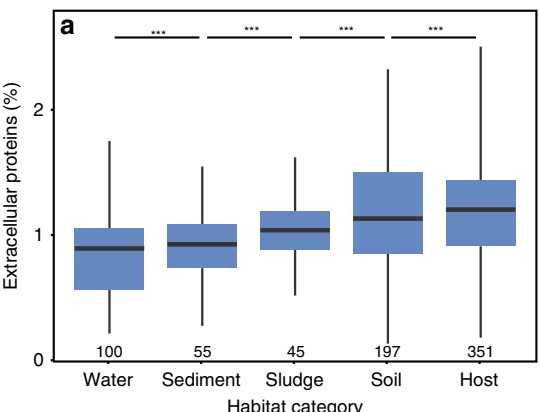
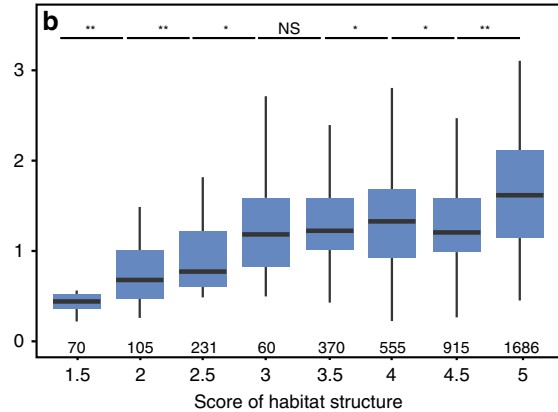

**Fig. 2 Relationship between the percentage of extracellular proteins and the habitat.** Boxplots represent the percentage of genes encoding extracellular proteins per genome in the different habitat categories (**a**) and in terms of habitat structure score (**b**). Boxes delimit the first and third quartile and whiskers represent 1.5 times the interquartile range. **a** Distributions were calculated using bacteria found only in one type of habitat (to avoid counting multiple times bacteria present in many environments). Results of Tukey HSD tests: NS, $P > 0.05$, $*P < 0.05$, $**P < 0.01$, $***P < 0.001$. **b** All bacteria had a habitat structure score that is computed to account for the diversity of habitats where they can be found. For easier visualization, and comparison, the data was binned. Number of genomes included on each bin are indicated below the $X$-axis. Comparison between bins was analysed using pairwise two-ways Wilcoxon test (corrected using the Benjamini-Hochberg method). NS, $P > 0.05$; $*P < 0.05$; $**P < 0.01$; $***P < 0.001$. Supplementary Fig. 4 shows the scatterplot associated with the non-binned data (where $\rho = 0.21$, $P_{MCMC} = 0.0012$). Source data are provided as a Source Data file.

(siderophores)[30]. We searched for the bacterial species of the genomes on these 16S rRNA datasets to link the environmental and the genomic data on genes encoding extracellular proteins (see 16S rRNA data and taxonomic classification). The 16S rRNA genes of the bacteria included in the analysis were used to build a phylogenetic tree that allowed to control for the phylogenetic structure of the data using MCMCglmm[31]. We found different frequencies of genes encoding extracellular proteins across environmental categories (Fig. 2a). We then produced an average habitat structural score for each OTU. For this, we divided the 16S datasets in environmental categories. Each category was given a score, following a previous work (Kummerli et al.[30]), where poorly structured environments have low score (from freshwater = 1) and very structured environments have high score (soil = 4, host = 5). The choice of host as the highest score is motivated by high habitat fragmentation of host-associated habitats. Results are qualitatively equivalent if the host category if removed (Supplementary Table 4). The score for a species is a function of the structural score of the habitats where the genome OTU could be found (see classification of habitats in the Methods section). The average frequency of genes encoding extracellular proteins is positively correlated with the habitat structural score ($\rho = 0.21$, $P_{MCMC} < 0.01$, Fig. 2b, Supplementary Fig. 4 for non-binned data). We made the same analysis with the bacteria found in one single habitat category (136 species, 15% of the total), to control for the possibly confounding effect of bacteria present in multiple habitats, and found qualitatively similar results ($\rho = 0.32$, $P_{MCMC} < 0.05$). This shows that bacteria encoding more types of extracellular proteins are more frequent in communities present in more structured habitats.

In poorly structured habitats, extracellular proteins remain close to the producer for a short period of time if their diffusion rates are high. In these conditions, lower rates of diffusion favor the producers, because they increase the return on the investment of producing the protein[8]. To test the hypothesis that diffusion length evolves in response to habitat structure, we computed protein diffusion lengths using the Stokes-Sutherland-Einstein equation with parameters given by the optimum growth temperature of the bacteria and the molecular weight of the protein. This analysis revealed lower diffusion lengths of the

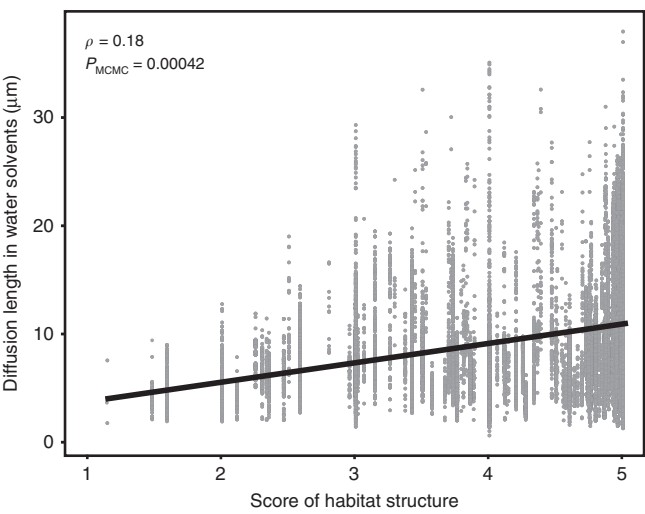

**Fig. 3 Diffusion length of extracellular proteins in function of the score of environmental structure.** Source data are provided as a Source Data file.

extracellular proteins in less structured habitats ($\rho = 0.18$, $P_{MCMC} = 0.00042$) (Fig. 3). Similar results were obtained by using only bacteria that were identified in a single category of habitats ($\rho = 0.19$, $P_{MCMC} = 0.00012$), and when we made the analysis per functional subcategory (Supplementary Fig. 8). These results show that bacteria identified in unstructured habitats have fewer genes encoding extracellular proteins and that the latter have lower diffusion lengths. This suggests an adaptation of the repertoire (and possibly the sequence and structure) of extracellular proteins to the habitats of the bacteria.

**Repertoires of extracellular proteins vary with community diversity.** One might expect bacteria living in highly diverse communities to encode fewer extracellular proteins because the probability that other bacteria profit from the public good is higher. Also, more diverse communities tend to have higher functional richness and this may decrease the need for the

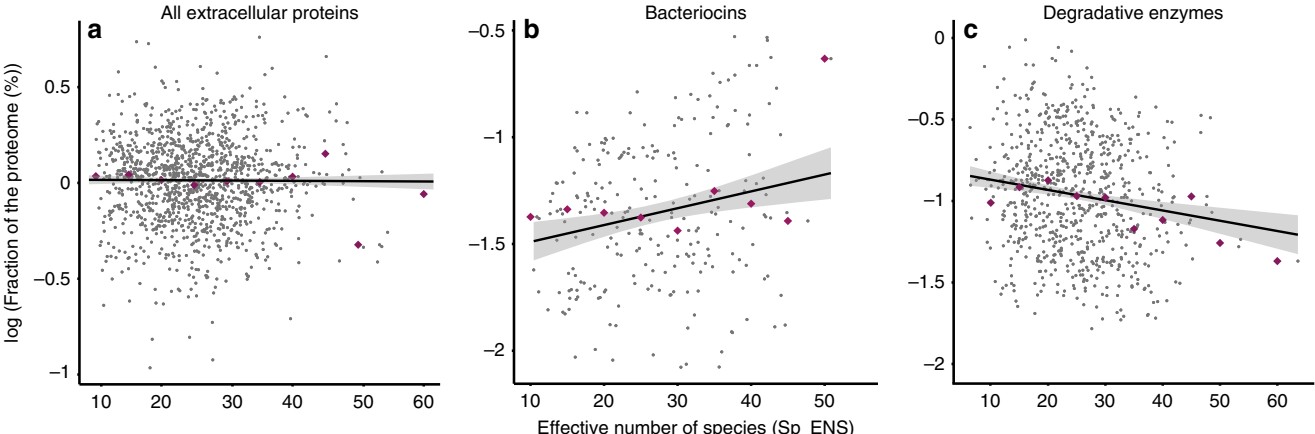

**Fig. 4 Frequency of genes encoding extracellular proteins (in *Y* axis) and its association with the alpha diversity (measured as the effective number of species, in *X* axis).** Dots in blue represent individual species, while purple diamonds represent the median values of each percentile. The shaded black line represents the linear regression associated to each distribution. Each panel represents one specific dataset: **a** Extracellular proteins (Spearman correlation $\rho = 0.002$, $P = 0.482$); **b** Extracellular bacteriocins ($\rho = 0.18$, $P_{MCMC} = 0.006$, regression statistics: $a = 0.362$, $b = 0.08$, $r^2 = 0.048$, $P < 0.05$); **c** Degradative enzymes ($\rho = -0.16$, $P_{MCMC} = 0.0003$, regression statistics: $a = -0.267$, $b = 0.564$, $r^2 = 0.063$, $P < 0.05$). Source data are provided as a Source Data file.

production of extracellular proteins. On the other hand, more diverse communities open the possibility of more diverse ecological interactions and these often require extracellular proteins. This is the case of many antagonistic interactions that take place through the production of extracellular toxins. Our analysis revealed no significant correlation between the average diversity of the habitats of a species - measured by the effective number of species—and the frequency of genes encoding extracellular proteins ($\rho = 0.007$, $P = 0.7905$, Fig. 4a). Superficially, this suggests lack of association between the frequency of genes encoding extracellular proteins in a genome and the diversity of the community.

We tested if this lack of correlation could result from confounding effects associated with the diversity of functions of extracellular proteins. We selected two groups of extracellular proteins that we could class reliably: bacteriocins (involved in antagonistic interactions) and degradative enzymes (involved in nutrient acquisition). It should be noted that they correspond to a small fraction of the entire dataset (8.47%). We found 5166 genes encoding for bacteriocins, of which 1037 were classified as extracellular—it's well known that many bacteriocins require cell death for dispersion in the environment[32]—an average of 1.96 per species. The frequency of genes encoding extracellular bacteriocins was higher in bacteria of very diverse communities (Fig. 4b). The same result was obtained for the analysis of the set of all bacteriocins ($\rho = 0.20$, $P_{MCMC} = 0.0012$). We identified 8280 genes coding for extracellular degradative enzymes, and their frequency decreased with the average diversity of the communities where the bacteria could be identified ($\rho = -0.16$, $P_{MCMC} = 0.0003$, Fig. 4c). We evaluated whether these results could be associated with an uneven distribution of bacteria across the values of alpha-diversity. Accordingly, we performed the same analysis 1000 times, using an equal number of 16S rRNA datasets for each habitat category. In all cases we obtained the same result ($P_{MCMC} < 0.05$). These results suggest that the frequency of extracellular proteins is affected by community diversity. The sign of this effect depends on the ecological role of the protein.

**Cost of extracellular proteins depends on diversity and structure.** Extracellular proteins tend to use amino acids that are less expensive to produce than the proteins classed as cytoplasmic (Fig. 5a), confirming previous reports[13]. Bacteria living in highly

diverse communities should be under stronger selection to lower protein biosynthesis cost, because exploitation is more likely in more diverse communities. Indeed, the average amino acid cost of extracellular proteins was lower in communities with high alpha diversity (Fig. 5b). These results suggest that bacteria in very diverse communities endure stronger selection for low-cost extracellular proteins.

We then hypothesized that selection to lower the cost of extracellular proteins could be stronger in well-mixed than in highly structured habitats. This is because direct reward for the production of these proteins is expected to be lower when proteins diffuse faster, making the cooperative process intrinsically more expensive. However, we did not observe an association between the cost and the habitat structure score ($\rho = 0.01$, $P_{MCMC} = 0.88$). To inquire on the reasons of this result, we accounted for the possibility that bacteria living in many different habitats may produce different extracellular proteins in different habitats. To remove this confounding effect, we did the same analysis using only the genomes of bacteria found in one category of habitat. In this case, we observed a positive association between the degree of structure of the habitat and the average biosynthesis cost of the extracellular proteins encoded in the genomes of the species found there ($\rho = 0.24$, $P_{MCMC} < 0.05$, Supplementary fig. 5). Intriguingly, this analysis revealed that the cost of extracellular proteins is lowest in soil bacteria. To detail this observation, we computed the distribution of genomic G+C content in each category of habitat. This revealed that bacterial genomes' G+C content is not significantly different across habitats (average 48%, $P > 0.05$ for all pairs, Tukey HSD test), except for soil bacteria, where it's higher (56%, $P < 0.01$ same test). This was observed previously and remains without explanation[33]. High G+C genomes, because of the structure of the universal genetic code tend to over-represent less expensive amino acids[34], and explain the lower cost of soil bacteria proteins. The genomic G+C content affects both extracellular and non-extracellular proteins and is not specifically associated with the object of this study. Removal of soil bacteria shows a stronger correlation between the biosynthesis cost and the degree of habitat structure ($\rho = 0.38$, $P_{MCMC} < 0.01$).

**Generalists encode more extracellular proteins.** Bacteria inhabiting more diverse sets of habitats may require a larger number

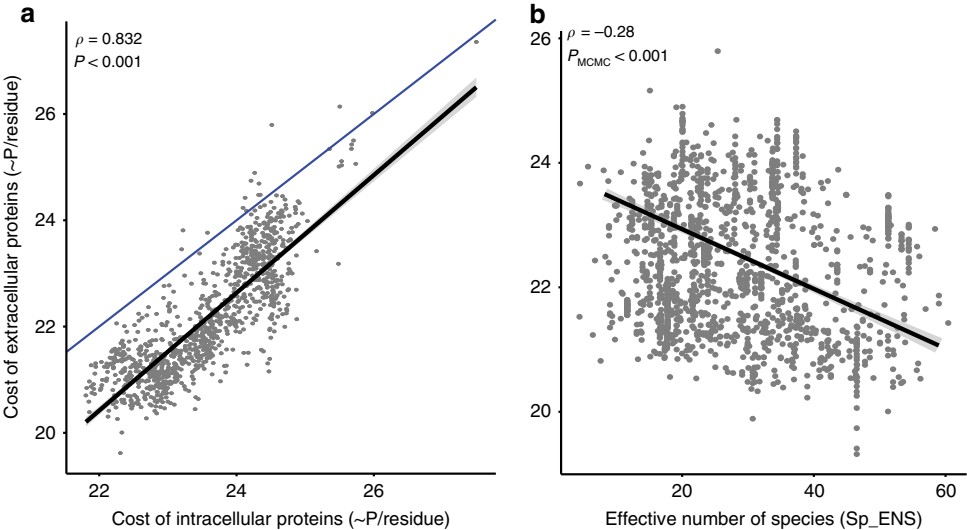

**Fig. 5 Relatioship between the biosynthesis cost and the genomic and environmental features. a** Comparison between the average biosynthesis cost per residue of extracellular and intracellular proteins. The two variables are correlated, but on average the extracellular proteins have less costly amino acids (Wilcoxon Rank test, $P < 10^{-10}$). The blue line represents the identity line while the black line indicates the linear regression (statistics: $a = 1.229$, $b = -6.623$, $R^2 = 0.683$, $P < 10^{-16}$). **b** Association between the average amino acid cost of extracellular proteins and the average Effective number of Species across the samples where a species was identified, regression statistics: $a = -1.449$, $b = 71.589$, $R^2 = 0.103$, $P < 10^{-16}$). Points represent individual species. Source data are provided as a Source Data file.

of genes encoding extracellular proteins to cope with the different characteristics of these habitats. To evaluate this hypothesis, we counted the number of different habitat sub-categories where each bacterial species could be found. We observed a significant correlation between this number and the frequency of genes encoding extracellular proteins in the genome ($\rho = 0.27$, $P_{MCMC} = 0.0017$). Hence, bacteria found in more diverse habitats tend to have more genes encoding extracellular proteins.

We detailed this result using three independent analyses. First, we compared the extremes of our distribution to control for uncertainty in the identification of habitats. Bacteria identified in a broad range of environments—generalists—have more genes encoding secreted proteins than bacteria found in few habitats—specialists (Wilcoxon rank test, $P < 0.01$, Fig. 6a). Second, we controlled for the possibility that small 16S rRNA samples could result in the classification of some bacteria present at low abundance as missing, which would tend to class them as specialists. We thus manually classed as specialists the bacteria known to be host-specific. This includes obligatory symbionts, such as intracellular pathogens and mutualists. The frequency of genes encoding extracellular proteins is greater in generalists than in host-specialists, even when accounting for genome size (same test, $P = 0.008$, Supplementary Fig. 6). Third, we compared generalists and specialists only for bacteria that are associated to hosts. We found similar qualitative results (same test, $P = 1.3 \times 10^{-4}$, Fig. 6b). All these results suggest a positive relationship between the number of habitats where a bacterium can be found and the frequency of genes encoding extracellular proteins in its genome.

### Discussion

We have shown that the frequency of extracellular proteins encoded by bacteria depends on habitat structure and community diversity. Some of the analyses underlying these conclusions showed a lot of variance that may have multiple causes. First, metagenomics and genomics data are heterogeneous. To tackle this, we used many controls (dataset composition, phylogenetic assignation, functional categories, phylogeny, genome size, and G+C), resampling, and literature (to identify specialists). Second,

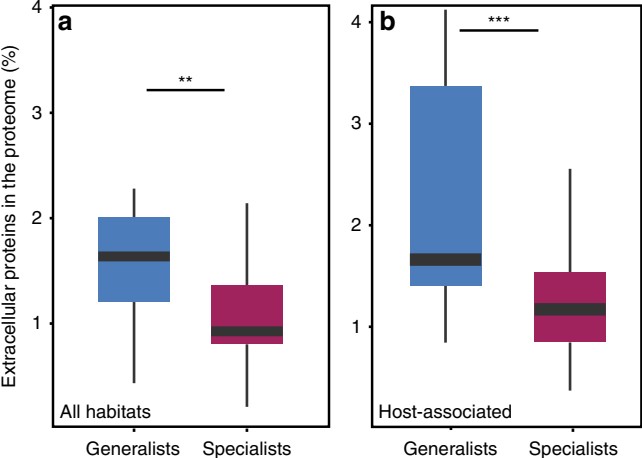

**Fig. 6 Association between the frequency of genes encoding extracellular proteins and the classification of bacteria in terms of the range of habitats where they could be identified.** The left panel indicates the results for all environments (sets **a** and **c**, described in Supplementary Information), and the right panel for host-associated habitats (sets **b** and **d**). Wilcoxon-Rank test. **$P < 0.01$; ***$P < 0.001$. Source data are provided as a Source Data file.

some species assignments may be inaccurate because 16S discriminates poorly between closely related species. Since close species tend to have similar genome sizes, gene repertoires, and lifestyles[35,36], this may have had little impact on our analysis. Third, identification of extracellular proteins may be inaccurate. Secretion signals are poorly known in some clades, even though we removed the least studied phyla, and can't be reliably identified in metagenomes because genes are often truncated and secretion signals are at their edges[1]. The use of 16S rRNA datasets and complete genomes also allowed to introduce controls for the effects of phylogeny and host genome size (unknown for metagenomics contigs). Fourth, species absent from 16S samples are

necessarily ignored. For example, *Bordetella pertussis* was not identified in any of the samples. This is consistent with its very low prevalence (thanks to the existence of a vaccine), but it also implies that it could not be used in the analysis of host-association. Finally, improved measures of diversity, e.g., phylogenetic and functional diversity[37], will increase the power of future statistical analyses.

The existence of variability in biological processes is another source of dispersion in the data. Genome analyses reveal the presence of a gene, but do not indicate if or when it's expressed. They also do not reveal if there are regulatory mechanisms that decrease the cost of cooperation by favouring gene expression under conditions of low exploitation, e.g., expression activated by quorum-sensing[38]. Our analysis also assumed that habitat structure favors cooperation. While theory suggests this is often the case, habitat structure can also increase the relatedness between potential competitors, which disfavors cooperation and in extreme cases cancels the effect of kin selection[39], leading to no correlation between the frequency of extracellular proteins and habitat structure. Finally, we assume the existence of potential interactions mediated by the extracellular proteins between organisms present in the same sample. This may not be the case if densities are very low, if the habitat is very structured, or if the proteins have no role in the biology of certain species.

In spite of these limitations, we have clearly shown that a physico-chemical trait of extracellular proteins—diffusion length—depends on habitat structure. High diffusion length implicates that extracellular proteins rapidly diffuse away from the producer. Although this may be selected in certain situations, e.g., the killing of a multicellular predator, it will tend to be very costly in most situations because the density of the effector decreases with the cube of the distance to the producer. This contributes to explain why bacteria have developed so many mechanisms to deliver toxins and diverse effectors to other cells by direct contact[40]. Because proteins provide more direct rewards to the producers if they don't diffuse very quickly in the environment, one expects stronger selection for proteins with small diffusion length in bacteria present in poorly structured habitats. This fits our observations. An analogous result was observed in the study of smaller molecules: poorly diffusible siderophores are more frequent in species living in unstructured habitats[30].

The diversity of communities is also associated with another biochemical trait of proteins: their amino acid biosynthesis cost. All extracellular proteins are expected to strongly select for amino acids that are less costly to synthesize, relative to cytoplasmic proteins, because these amino acids cannot be recycled when the protein is degraded. Indeed, the usage of less expensive amino acids is higher in extracellular than in cytoplasmic proteins, as initially observed in *E. coli*[13], and later in other genomes[41]. Here, we extend these previous observations by showing that highly diverse communities over-represent proteins with less expensive amino acids relative to less diverse communities. Under Hamilton's rule[42], altruism evolves when the product of the benefit (b) and the relatedness of the individuals benefiting from altruism (r) is higher than the cost (c) of the process to the producer (i.e., when b.r > c). This provides an explanation for our results. When cooperative processes depend on the production of an extracellular protein, the cost of the protein shapes the cost of the cooperative process and thus its evolutionary stability. As a consequence, one expects stronger selection to lower the cost of extracellular proteins when the risk of exploitation is higher (more diverse communities).

The variation in the characteristics of proteins in function of community diversity and habitat structure can be explained by two different mechanisms. First, by natural selection of the events of gain and loss of genes encoding extracellular proteins. The gene repertoires of these proteins are known to change constantly by horizontal transfer and gene loss[13,16]. Our results suggest that the probability of retention of recently acquired genes encoding extra-cellular proteins may change in function of environmental traits—such as habitat structure and community diversity—because these affect the efficiency of selection for expensive and rapidly diffusing proteins. Bacteria in well-mixed habitats or diverse communities will require stronger natural selection to maintain certain types of extracellular proteins. Second, recently acquired protein sequences may evolve, by natural selection of mutations decreasing production costs and lowering diffusion lengths, to adapt the extracellular proteins to the species' habitat. Disentangling the relative importance of the two mechanisms, gene repertoire variation and protein sequence evolution, will require the analysis of within-species polymorphism of nucleotide sequences and of gene frequencies. This will require many strains for each species because these genes are present in a small number of strains of a species[16]. At the moment these are available in sufficient number only for very few species, mostly host-associated. The rapid growth of genome databases, including genomes assembled from metagenomes, will soon make such analyses possible.

The effect of community diversity on the frequency of genes encoding extracellular proteins seems to depend on the function of the extracellular protein. We observed that degradative enzymes are less frequent in more diverse communities, in line with the idea that bacteria growing in species-poor communities will more often invest in extracellular proteins allowing to scavenge the environment for resources. One could have envisaged the hypothesis that higher competition for nutrients in more diverse communities would result in a higher number of genes encoding degradative enzymes. We do not observe such an effect in our data, possibly because such competition may have a stronger effect on the level of expression of enzymes than on their functional diversity. In contrast, the frequency of extracellular bacteriocins does increase with community diversity. These results raise the hypothesis that host species-rich communities, like the gut lumen, have many genes encoding extracellular proteins involved in antagonistic interactions with other microbes. This antagonistic interaction could have a strong impact in the protective role of the native microbiota against the colonization by opportunistic pathogens[43].

The association between habitat structure and the frequency of extracellular proteins, explains that host-associated bacteria have some of the highest frequencies of genes encoding extracellular proteins. This fits previous studies showing that structured habitats favor the production of extracellular toxins[44], and the numerous observations that extracellular proteins are key virulence factors in most bacterial pathogens[45] and key components of mutualistic interactions[46]. Bacteria that shift between highly structured and poorly structured environments can cope with this using genetic regulation. For example, *Pseudomonas aeruginosa* decreases the production of extracellular proteases and toxins in unstructured habitats leading to a loss of virulence[47]. Bacteria may also evolve mechanisms to tackle directly the exploitation of extracellular proteins by non-producers or rapid protein diffusion in poorly structured habitats. For example, kin recognition using quorum-sensing allows to restrict the production of public goods to when the bacterium is surrounded by producers[48]. Other mechanisms include the privatization of the extracellular protein (the case of many beta-lactamases in Proteobacteria[49]), the restriction of the expression of extracellular components to appropriate environments[50], the niche construction leading to increased habitat structure favouring cooperation by the production of biofilms or phase separation through Type 6 Secretion killing[51], and the development of antagonistic interactions with

non-kin to deplete the local environment from cheaters[52]. In conclusion, Bacteria encoding many different extracellular proteins tend to be identified in structured habitats, but the others can still mitigate the cost of extracellular proteins by several means.

Whether host-associated or not, we found that generalists dedicate a larger fraction of their gene repertoires to extracellular proteins than specialists. This effect can be explained in several ways. Extracellular proteins may affect bacterial growth by facilitating the use of available nutrients. In this case, the genomes with a more diverse repertoire of extracellular proteins are those able to grow in a larger range of habitats. This is likely to affect the frequency of genes encoding proteases, lipases, or hemophores. This situation is analogous to that of small compounds secreted to scavenge iron[53,54]. Extracellular proteins also provide advantages during colonization when they clear the niche from competitors, contributing to explain why toxins are more frequent in bacteria of highly diverse communities. This in line with works showing that extracellular proteins are key factors on the range expansion of pathogens, by allowing them to adapt to new hosts or modify them to their benefit[55]. It's interesting to remark that bacteria able to produce broad-host range antibiotics, which are extracellular small molecules, tend to be generalists that inhabit the highly diverse communities of strongly structured habitats in the soil.

In this work, we have specifically focused on extracellular proteins and their potential role as public goods. We expect that many of our results will hold when applied to the study of other extracellular compounds when these are costly public goods. The larger implication of our work is that the gene repertoires of extracellular proteins of bacteria, which determine their ability to use and change their environment, depend on the characteristics of the habitat and the community. This may contribute to explain why some toxins are secreted directly into other cells using expensive protein secretion systems or just secreted to the extracellular space. It may also explain why the repertoires of extracellular proteins vary so widely across bacteria. Finally, it shows the importance of putting together biochemical (biosynthesis cost, diffusivity) and ecological information (habitat structure, community diversity) in an evolutionary context.

## Methods

**16S rRNA data and taxonomic classification**. We collected 11,649 16S rRNA amplification datasets from MG-RAST (last accessed: march 2015). The datasets had very diverse sizes, with an average size of 100,473 sequences per dataset (Supplementary data 1 and Supplementary Fig. 7). We removed those containing non-ribosomal data or less than 2000 sequences. For this, we queried the sequences of each set for similarities to a 16S rRNA profile using SSU-align v. 1.0.1[56] using the search algorithm and default parameters. SSU-align uses the conserved secondary structure and sequence of SSU rRNA to identify these sequences. A total of 2329 datasets containing less than 85% of sequences matching the 16S profile were regarded as having low quality and were discarded, leaving 9320 datasets for further analysis (Supplementary Fig. 8).

Sequences were aligned against the same structural profile with SSU-align using the search & global_align algorithm, in order to obtain multiple sequence alignments. The multiple alignments were trimmed using trimal v 1.4[57] using the automated1 algorithm. The resulting trimmed sequences were clustered into redundant groups of 99% identity using the uclust algorithm from usearch v. 9.0.2132[58]. A reference sequence was selected for each redundant group from each dataset, by selecting the closest sequence to the centroid coordinates. We then defined a global catalog of operational taxonomic units (OTU) for the whole dataset by clustering the reference sequences with usearch into reference OTUs of 97% identity. The reference sequences of each OTU were classified taxonomically against the RDP databases[59] at species level using blastn v. 2.2.26+ with a minimum e-value of $10^{-5}$ and a minimum coverage of the query 16S rRNA of 80%. We discarded the 38% of the reference sequences that could not be classed in a genus.

**Diversity indexes**. We used the expected number of species (ENS) as a metrics of the alpha diversity of each 16S rRNA dataset[60]. In order to calculate it, we first used the Shannon diversity index (H'):

$$H' = -\sum_{i=1}^{R} p_i \ln(p_i), \tag{1}$$

where $p_i$ is the relative frequency of a specific species in the dataset (the number of sequences associated with the species divided by the total number of sequences assigned to species), and $R$ is the number of datasets. Based on the Shannon diversity, we calculated the ENS as the exponential of the Shannon diversity:

$$\text{ENS} = e^{H'}. \tag{2}$$

Finally, we calculated the average values of ENS for the set of habitats where a species ($i$) was identified. This was defined as the average value of ENS across the samples ($j$) where it's present:

$$\text{Sp\_ENS}_i = \frac{\sum_{j=1}^{\text{Nd}_i} \text{ENS}_j}{\text{Nd}_i}, \tag{3}$$

where $\text{Nd}_i$ is the number of datasets where the species was found. In order to avoid biases associated with the over-representation of specific habitat categories, we made 1000 random samples of an equal number of datasets for each category, and calculated each diversity index as the mean of all measurements (the values can be found in Table S2).

**Classification of habitats**. The 9320 16S rRNA datasets were classified in terms of habitat using a previously defined nomenclature[36], in seven main categories (water, sediment, wastewater/sludge, soil, and host-associated), and 21 sub-categories (henceforth called sub-environments) (Supplementary Data 1 and Table 1).

These categories were divided in five broad groups of increasing habitat structure, and we assigned an integer ($h$) (from 1 to 5) to each one of them using the method of Kummerli et al.[30], to which we removed the distinction of freshwater and marine water, and added a category for Wastewater/Sludge. The rationale behind this classification is associated to the habitat fragmentation generated by its structure; unstructured habitats such as water are seen as a continuum, where any solute molecule has the capacitiy to diffuse freely according to its molecular mass, while highly structured habitats (including soil or host-associated), have physical barriers that create microenviroments, preventing the free distribution of molecules. The habitat structure score $h$ was given as follows: natural aquatic, including freshwater and marine environments ($h = 1$), sedimentary soils from aquatic environments ($h = 2$), wastewater/sludge ($h = 3$), soil ($h = 4$), and host-associated ($h = 5$).

Bacterial species were assigned a habitat structure score (habSS) according to the dataset (i) where they were identified (Supplementary data 3):

$$\text{habSS}_B = \sum_{i=1}^{n_B} \frac{h_i * c_{i,B}}{T_b}, \tag{4}$$

where $n_B$ is the number of 16S rRNA datasets where the bacterial species B was identified (see Genomic data, taxonomic classification and phylogenetic reconstruction on how species were assigned to OTUs), $c_{i,B}$ is the number of 16S rRNA reads assigned to species B in the dataset $i$, $T_b$ is the total number of reads in the datasets associated to species B, and $h_i$ is the value of the habitat structure score for sample $i$.

Finally, 17 species (1% of the total number) were not found in the 16S rRNA data and could not have a score computed from the data. The environmental structure score of these species was attributed from literature data. 10 out of these 17 species were characterized as obligatory symbionts, including species such as *Brucella melitensis, Chlamydia psittaci, Dyckeia dadantii, Mycobacterium tuberculosis, Treponema pallidum, Rickettsia felis,* or *Sulcia muelleri,* among others. The other seven species included bacteria isolated from soil (*Geobacter lovleyi, Pimelobacter simplex, Solibacillus sylvestris, Streptomyces avermitilis*), sea water (*Echinicola vietnamensis, Synecococcus sp. WH7803*.), or from highly specific environments, such is the case of *Kinecoccus radiotolerans,* a soil-associated bacterium isolated from nuclear waste-contaminated sediments[61].

**Identification of generalists and specialists**. The classification of species into generalists and specialists was performed as follows. We first counted the number of different habitat sub-categories where each bacterial species was identified. In order to account for the prevalence of each species in each habitat sub-category, we calculated the average frequency of each species in each sub-category. The final number of habitats ($E$) was then weighted by the relative frequency of the species in each habitat sub-category, using the formula:

$$E_i = \sum_{j=1}^{t} F_{ij}, \tag{5}$$

where $F_{i,j}$ stands for the average frequency of the species $i$ in the sub-category $j$.

From the previous equation, we defined four datasets: species in dataset A, henceforth called specialists, includes the species found in the first quantile of the distribution of the number of habitats ($E$). Within the specialists, Dataset B (host

specialists) was defined by those bacterial species in (A) known to have a strict relationship with hosts, according to the literature[61]. Dataset C, henceforth called generalists, includes the species found in the last quantile of the distribution. Finally, Dataset D (host-generalists) included the species in C that are found in hosts. We also identified 136 specialist species (889 genomes, 15%) which were only found in one habitat, and which were used as a validation dataset.

### Genomic data, taxonomic classification, and phylogenetic reconstruction.

We retrieved all the 5775 completely assembled genomes, including 4794 plasmids, available in GenBank RefSeq (Supplementary data 4, last accessed November 2016). We made two filters on this dataset, one on the number of taxa per phyla, another associated with the 16S mapping procedure. First, to have enough statistical signal, we restricted our analysis to phyla with at least 50 sequenced genomes, resulting in 3922 genomes. We then matched the 16S of the genomes to those of the 16S datasets. For each genome, a representative 16S sequence was extracted from the genome and was aligned using SSU-align with the bacterial structural model. The newly aligned representative sequences were manually trimmed at both edges to adjust the size to the 16S rRNA dataset[62]. The sequences were assigned to a species level using the RDP classifier from the RDP database. The assignation was performed using the Lowest Common Ancestor algorithm, which identifies the lowest convergent taxonomic assignation for all the significant hits of each alignment[63]. Genomes with ambiguous species classification, i.e., classed in several different species with similar confidence levels, where discarded from further analyses, leaving a total of 1104 species and 3817 genomes.

Since the results are controlled for phylogenetic structure, there is no need to remove further phylogenetic redundancy from the datasets. The 16S rRNA alignment containing the representative sequences of the remaining genomes was used to build a phylogenetic tree using IQ-tree[64], which identified the substitution model GTR+F+I+G4 as the best (based on the Bayesian Information Criterion). Thousand Bootstrap trees were constructed to determine the topology support at each node.

### Identification of extracellular proteins, taxonomic, and functional classification.

We predicted the sub-cellular location of the proteins encoded in the genomes included in the analysis (see Genomic data, taxonomic classification, and phylogenetic reconstruction) using PSORTB v 3.1[29]. The PSORTB model was selected based on the species' monoderm/diderm classification (taken from the literature)[61]. Only proteins classified as "extracellular" by PSORTB and lacking transmembrane domains where considered in our study. Proteins not matching these criteria were discarded. When more than one genome was available per species, we computed the average number of proteins per genome for that location (Supplementary data 5). Extracellular proteins were functionally classified by searching for sequence similarity, using HMMsearch from HMMer v.3.1.2b[65], in the eggNOG v. 4.5 database[66]. We only considered hits with an $e$-value $\leq 10^{-5}$ and more than 50% similarity. Since different HMMs may be associated to the same functional category in different taxa, we kept the functional annotation of the best hit when more than half of the hits were associated to that same category (otherwise it was marked unknown).

Three functional categories were explored more carefully. First, we characterized the repertoire of extracellular bacteriocins. To do so, we searched for similarities to the extracellular proteins in the two bacteriocin databases Bagel and Bactibase[67,68] using HMMer. We kept the hits with an $e$-value < 0.05 and more than 50% coverage of the query sequence (Supplementary Table 2). Second, we identified the extracellular proteins with a degradative activity. We selected enzymatic activities often associated to the extracellular environment: amidase, amylase, cellulase, chitinase, dipeptidase, glycosyl hydrolase, invertase, inulinase, keratinase, and pectinase[69]. For each degradative enzyme, we collected all previously validated bacterial protein candidates by searching for specific keywords in Uniprot[70]. We clustered them using usearch with the "cluster_smallmem" algorithm at 70% identity. We aligned the sequences of each cluster using mafft v.7 with the local pairwise alignment option and a maximum 1000 iterations ("linsi" option)[71]. The resulting multiple alignments were used to build protein HMM profiles using hmmbuild from HMMer. HMM profiles were queried against the extracellular proteins previously predicted. Hits with more than 40% identity and less than 20% difference in length for the smallest of either the protein or profile where kept, and the best hit was used to classify them (Supplementary Table 2).

### Quantification of chemical properties of extracellular proteins.

We estimated the diffusion length of a protein ($d$), as in Bard and Faulkner[72]:

$$d = \sqrt{Dt}, \tag{6}$$

where $D$ is the diffusion coefficient and $t$ the time in seconds.

In aqueous solution, $D$ can be estimated using the Stokes-Sutherland-Einstein equation as proposed by Kalwarczyk et al.[73]:

$$D = \frac{kT}{6\pi\eta r_p}, \tag{7}$$

where $k$ is Boltzmann's constant, $T$ is the temperature of the solvent, and $\eta$ is the viscosity on the solvent. Finally, $r_p$ is the hydrodynamic radius of a protein, which

can be calculated according to its molecular weight ($M_w$):

$$r_p = 0.0515 M_w^{0.392}, \tag{8}$$

We computed the diffusion coefficient for cytoplasm and water, using viscosity values from the literature[74,75]. Given the lack of data on the temperatures associated to the environments where the different species could be isolated, we decided to make the simplification that bacteria are in habitats close to their optimal growth temperature ($T_o$). For each complete genome included in the analysis, we calculated $T_o$ as in Vieira-Silva et al.[76].

Extracellular proteins are predicted to be costly because their constituents cannot be recycled easily[13]. To evaluate the effect of the metabolic cost in the environmental distribution, we calculated the biosynthetic cost (in ATP equivalents) per amino-acid for each protein, suing a previously published method[77] (Supplementary Data 6).

### Statistical analyses.

We evaluated the phylogenetic effect of the association between the diffusion coefficient of extracellular proteins and the habitat structure, using the R package MCMCglmm[31]. We used the phylogenetic tree of the 16S rRNA sequences as a random factor in the model, according to the package guidelines, with a variance limit to 1. For each analysis, we used 300,000 model iterations with a starting burn-out phase of 50,000, sampling every 100 iterations. From the posterior distributions obtained in this Bayesian analysis, we extracted the $P_{MCMC}$ (used here as $P$-value). A low (non-significant, alpha = 0.05) $P_{MCMC}$ indicates that we cannot exclude the possibility that associations are simply due to phylogenetic structure.

Given the bias towards host-associated habitat datasets in the public repositories, we controlled each correlation test by performing 1000 random samplings of an even distribution of habitats. In all cases, we observed equivalent results for all replicates.

### Reporting summary.

Further information on research design is available in the Nature Research Reporting Summary linked to this article.

## Data availability

All datasets (bacterial genomes and 16S rRNA) used in this work are publicly available. To know more about the repositories and reference IDs, please refer to Methods and Supplementary dataset 1.

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

## Acknowledgements

This work was supported by a grant from the European Research Council [EVOMOBI-LOME, 281605 to E.P.C.R.], and by the INCEPTION project [PIA/ANR-16-CONV-0005 to Pasteur Institute]. We thank comments and suggestions from Carmen Bessa Gomes, Sophie Abby, Jorge Sousa, Philip Engel, Germán Bonilla-Rosso, and Kirsten Ellegaard.

## Author contributions

E.P.C.R. and M.G.G. designed the study. M.G.G. produced and managed the data. M.G.G. made the analyses with contributions from E.P.C.R. M.G.G. and E.P.C.R. wrote the manuscript.

## Competing interests

The authors declare no competing interests.
