## [Peer Review File · Nature Communications]

Reviewers' Comments:

Reviewer #1:

Remarks to the Author:

The authors integrate multiple sources of microbial genomic data to illustrate a number of exciting inter-specific relationships that are broadly consistent with current social evolution theory: (1) secreted proteins are enriched in species found in structured environments; (2) degradation enzymes enriched in low diversity envs, toxins in high diversity environments; (3) secreted protein costs are minimized in less structured and more diverse environments; (4) secreted proteins enriched in generalists.

This is a very ambitious and fun paper to read – full of interesting results, and yes – lots to argue with. The authors recognise in the discussion that their datasets are noisy and therefore estimated effects are likely to be small. I'm more concerned about some of the calculated metrics, which are introduced with very little discussion and no critique–

1) Habitat structure score

On a scale of 1 (mixed) to 5 (structured), why does soil score 4 and host-associated score 5? There is no justification for this that I could find, and certainly my apriori is that soil is more structured than a host lumen. I imagine the results would hold with a 3 scale water – sediment/sludge – soil / host, which would be a little more conservative. Even then I'm still wondering, is sludge more structured than living on a host, on the micron-scale view of a microbe? There really needs to be a principled way of developing and justifying this metric.

2) diversity score

there's a bit more detail on the math behind the calcs here (in the SI), but there's a lack of biological justification. I'm concerned about the spatial scale of diversity. More specifically, I'm concerned you're assuming that all bugs within a single environmental sample that underwent 16S profiling are potentially interacting. This is not necessarily the case, and is worth some consideration. I imagine this is the best effort that can be done to address this question, but this limitation should be recognised along with the potential to impact the results and interpretation.

Other major concerns

3) Generalism -> secretome section

The authors should take a look at <https://www.nature.com/articles/ncomms5594> which covers much the same ground as this section. Certainly the current MS covers much more besides, so this is not a concern for potential acceptance in a high profile journal.

Minor concerns

Line 75: how do you square this 6% figure with Fig S1 where mode is 1% ish?

Fig 1: hard to read black numbers against dark purple background. Purple scaling - % of what?

Line 165-167: had a look at the table and there's some dodgy calls for host association at least – eg H influenzae, B pertussis and P aeruginosa are classed as not host associated. What human 16S samples were you using to make these calls? It is understandable that automated analyses of large datasets will produce errors of this kind, but I think worth flagging this class of error in the discussion to illustrate the 'noise' issue.

Line 395-399: bit of a non-sequitur here. Authors begin with virulence, but then switch to refs 45-46 which refer to bacteriocins. These kill other bacteria, but are not virulence factors from host perspective.

Fig S2: 3 colors not explained clearly

Reviewer #2:

Remarks to the Author:

Bacteria secrete different classes of compounds to interact with their abiotic and biotic environment. Such compounds include toxins to fight competitors, enzymes to digest food, and secondary metabolites to form biofilms, scavenge metals and swarm on surfaces. There has been tremendous interest in these secreted compounds because they fulfil critical functions for the bacteria themselves, but also for the community as a whole. The latter is the case because secreted compounds can potentially be shared as public goods between cells and thus foster social interactions between them.

While there is an enormous body of experimental work on single-species/single-trait systems in the laboratory, we still know very little on the ecology and evolution of these secreted compounds. In other words, the ecological conditions (abiotic and biotic) that select for compound secretion in natural settings are poorly examined.

The study by Garcia-Garcera and Rocha tackles this issue. The authors conducted a comparative analysis using genomic data (i) to infer the type and abundance of secreted compounds among bacteria and (ii) to quantify community diversity. They further collected data on the habitats the identified species populate. Using phylogeny-controlled analysis, they found that (a) species living in more structured environments secrete more compounds; (b) the diffusion length of secreted compounds is increased in more structured habitats; (c) community diversity correlates with the secretome size, but (as expected) differently for toxins vs. digestive enzymes; (d) there is evidence for selection for costs optimisation; and (e) generalists secrete more compounds than specialists.

The paper by Garcia-Garcera and Rocha is very well written and easy to follow. The topic is timely, highly relevant, and the insights provided represent a key step towards a better understanding of the evolution and ecology of secreted compounds and community interactions in bacteria. In brief, this is an outstanding paper.

Although, I am very positive I still have a few points I would like to see addressed:

Major comments:

(1) The authors should be more careful in explaining the chemistry and biology of secreted compounds. The current notion in the manuscript is "secreted compounds = proteins = public good". There are several complications, which are currently not addressed in the paper:

a) not everything that is secreted is necessarily a public good. Secreted molecules can be embedded in the membrane of the producer (e.g. biosurfactants) and might only partly be shared with other cells. This fact should be explained and strong statements like 'extracellular proteins are public goods' should be revised.

b) some secreted compounds can only be shared between cells that have a matching receptor for

uptake (e.g. siderophores). Again, this means that such secreted compounds are not necessarily an accessible public good for everyone in the community. This must be explained.

c) not all secreted compounds are proteins. In fact, a high proportion of them are secondary metabolites, where only the machinery for metabolite synthesis is encoded in the genome but not the compound itself. Secondary metabolites include toxins, antibiotics, biosurfactants, siderophores, biofilm matrix components, etc. This important class of secreted compounds is not clearly mentioned in the paper and it is also not clear whether they were included in the analysis. Presumably not. But this must be explained and stated very clearly.

d) with regard to the points above, it must be very clearly stated that the findings of the paper hold for secreted proteins only and not for secreted compounds in general. The dynamics and patterns for secondary metabolites might be quite different. The only other paper that pursued a similar approach is Kummerli et al. 2014 *Ecol Lett.* (on siderophores). The authors should compare their results to this earlier study and discuss generalities and differences. This would greatly help to obtain a more general understanding of the ecology and evolution of secreted compounds.

(2) In my opinion the general statement that secreted compounds are predominantly encoded on plasmids is wrong. The authors studied proteins, and for this class of secreted compounds the statement might be correct. Within this class, there is a heavy bias towards toxin-antitoxin systems, which are often encoded on plasmids. However, most (if not all) secreted secondary metabolites are encoded on the chromosome. There is a simple reason for this: most secondary metabolites are quorum-sensing controlled and this type of complex regulatory network requires a stable genetic architecture (i.e. the chromosome). This part of bacterial biology is completely neglected in all the papers on 'social traits being enriched on plasmids'. Careful phrasing is required to clarify that the pattern only holds for secreted proteins (and presumably only for toxins).

(3) This brings me to my last point. Currently, the authors combine different types of secreted proteins (enzymes and toxins) in most of their analyses. However, I suspect associations to vary between environmental factors and enzymes (providing a cooperative benefit to others) versus toxins (targeting competitors). This is clearly shown in Fig. 4. But what about the other analyses? For instance, it makes sense to evolve diffusible molecules in a structured habitat, as molecules can be shared between nearby relatives. Conversely, toxin production makes little sense in this context as they would only target resistant relatives. There is theory showing that toxins should be more diffusible than beneficial public goods and provide highest benefits at intermediate spatial structure (e.g. Inglis et al. 2009 *PNAS*, Wechsler et al. 2019 *JEB*). I recommend to split all analysis in enzymes vs. toxins as it was done in Figure 4. Introducing a color code for the two classes in the figures might suffice, and report differences in stats if there are any.

Minor comments:

line 35. Start with "We found" to delineate the general intro from the new findings.

line 39. Unclear what is scavenged.

line 84. Consider rephrasing. Currently it reads as if proteins pick amino acids from a collection for their assembly.

line 122 and other places. '16S' should always be spelled out as '16S rRNA'

line 153. Remove hyphens 'carbohydrates' and 'amino acids'

line 163. Statements like this are problematic as they only hold for cooperative proteins, but not for toxins (see major comment 3).

line 209. reword 'bacteria's habitat'

line 308. replace 'trends' by 'effects', as this relationship is significant and not just a trend.

lines 346-349. Confusing paragraph. Previously, the authors said that spatial structure favours cooperation, whereas here they state the opposite. I understand the point they want to make, but it might confuse non-specialist readers. Rephrase to something like 'although spatial structure is thought to favor cooperation, it can also have negative effects ...'

lines 376-378. I have some doubts (see major comment 2). I believe that this is only true for toxins.

lines 401-403. This is one reason why a healthy microbiome can fight off pathogens. This could be mentioned in this context.

Signed: Rolf Kümmerli

Reviewer #3:

Remarks to the Author:

Summary

This paper investigates the effect of community diversity and habitat structure on the repertoire of extracellular proteins in bacteria, with particular focus on extracellular protein frequency and cost. Extracellular protein repertoires are interesting and important as bacteria use them to interact with their environment, including social interactions with other bacteria. The authors use genomic data and 16S environmental datasets to study natural microbial populations.

The main findings are: (i) bacteria living in more structured habitats encode more extracellular proteins; (ii) the effect of community diversity depends on extracellular protein function: proteins involved with antagonistic interactions (bacteriocins) are more abundant in more diverse communities, whereas proteins involved with nutrient acquisition (degradative enzymes) are more abundant in less diverse communities; (iii) extracellular proteins are less costly in more diverse communities and have lower diffusivity in less structured habitats; (iv) bacterial generalists encode more extracellular proteins than specialists. Overall, the results demonstrate the importance of ecological factors in shaping the evolution of bacterial extracellular gene repertoires.

General comments

This is a strong manuscript with multiple novel and interesting findings. It would be of broad interest to microbiologists and social evolutionary biologists. The authors clearly state their results and the analyses conducted are thorough, using a large genomic sample size. Overall, this paper is a significant advance in our understanding of broad-scale ecological effects on extracellular protein repertoires in bacteria. Some improvements for the manuscript are detailed in the specific comments below.

Specific comments

1. The genome selection process is not explicitly clear from the manuscript (or supplementary material), how was this conducted?
2. Line 60/61 (and elsewhere) - "Extracellular proteins are public goods that can benefit...". This is not strictly correct, as not all extracellular proteins are public goods. I recommend the authors state that extracellular proteins can act as public goods.
3. Lines 214-220 – further alternative hypotheses should be considered when discussing the possible effect of community diversity on extracellular protein frequency. For example, one could hypothesise an increased number of degradative enzymes with increasing community diversity, to increase competitive ability during indirect competition for nutrients.
4. Lines 260-261 is lower protein biosynthesis only explained by higher likelihood of exploitation in diverse communities, or could it also be that a lower diversity of secreted protein production is required if each species produces a few that all can then use?
5. Figure 4. - I understand why the authors have chosen bacteriocins and degradative enzymes to represent antagonistic and cooperative proteins, respectively. However, to further support the authors' data interpretation of Fig 4, results for other public good groups such as nutrient acquisition enzymes would be useful, as degradative enzymes constitute a very broad group of proteins. Did the authors conduct this community diversity analysis on any other classes of extracellular proteins?
6. The discussion is currently quite disjointed. This could be improved by repositioning or reducing the extensive justification for the use of genomic data and the reasons for data dispersion. The discussion also ends abruptly and would be improved if the authors more thoroughly considered the boarder implications of their results.
7. Fig S3 – is there an explanation for why extracellular proteins from 'habitat structure 4' have such a significantly low amino acid cost that contradicts the general trend? Is there anything noticeably different about the genomes in this group?

Answers to reviewers – First round

Reviewer #1:

The authors integrate multiple sources of microbial genomic data to illustrate a number of exciting inter-specific relationships that are broadly consistent with current social evolution theory: (1) secreted proteins are enriched in species found in structured environments; (2) degradation enzymes enriched in low diversity envs, toxins in high diversity environments; (3) secreted protein costs are minimized in less structured and more diverse environments; (4) secreted proteins enriched in generalists.

This is a very ambitious and fun paper to read – full of interesting results, and yes – lots to argue with. The authors recognise in the discussion that their datasets are noisy and therefore estimated effects are likely to be small. I'm more concerned about some of the calculated metrics, which are introduced with very little discussion and no critique–

Answer: We thank the reviewer for the positive assessment of the manuscript.

1) Habitat structure score.

On a scale of 1 (mixed) to 5 (structured), why does soil score 4 and host-associated score 5? There is no justification for this that I could find, and certainly my apriori is that soil is more structured than a host lumen. I imagine the results would hold with a 3 scale water – sediment/sludge – soil / host, which would be a little more conservative. Even then I'm still wondering, is sludge more structured than living on a host, on the micron-scale view of a microbe? There really needs to be a principled way of developing and justifying this metric.

Answer 1.1: The measure we used in our work was proposed and used by other authors in a similar context (see Kummerli, Ecol Lett, 14, who is also reviewer #2). This explains why we made (admittedly too) little effort in explaining it. The rationale for putting host as one with higher value than soil has to do with habitat fragmentation that is much higher in host-associated habitats. Fragmentation will diminish diffusion of genotypes (and extracellular proteins) in the population. However, we do understand the reviewer's concern about the habitat structure score in what relates to the host-associated category.

Action: We added some text to explain the logic of the categorization (results and methods sections) and some text to add some critique (discussion). We also re-analysed the key results using a structural score where host datasets were removed from the data. The key results remain unchanged (new Table S10).

2) diversity score

there's a bit more detail on the math behind the calcs here (in the SI), but there's a lack of biological justification. I'm concerned about the spatial scale of diversity. More specifically, I'm concerned you're assuming that all bugs within a single environmental sample that underwent 16S profiling are potentially interacting. This is not necessarily the case, and is worth some consideration. I imagine this is the best effort that can be done to address this question, but this limitation should be recognised along with the potential to impact the results and interpretation.

Answer 1.2: We do agree that within a sample the bacteria may not interact. This will be especially true if densities are low or if the extracellular protein has no role in the biology of some bacteria. As stated by the reviewer there isn't much we can do about it, because there is no idea of the physical distance between the different organisms nor

the structure of the sample (beyond the classification of environments that we have made). If few of the potential interactions are actual, this would decrease the power of our tests, but we see no reason for it to bias our results. This may justify the large variance observed in certain analyses.

Action: We have added a sentence about this limitation on the discussion.

Other major concerns

3) Generalism -> secretome section

The authors should take a look at <https://www.nature.com/articles/ncomms5594> which covers much the same ground as this section. Certainly the current MS covers much more besides, so this is not a concern for potential acceptance in a high profile journal.

Answer 1.3: We thank the reviewer for pointing out this article that we knew, but forgot to cite. We also found the excellent McNally, Nat Comm, 2017 after this paper was submitted.

Action: We have included these works in our discussion.

Minor concerns

Line 75: how do you square this 6% figure with Fig S1 where mode is 1% ish?

Answer 1.4: The two numbers are compatible because one is the frequency of the set of gene families in an average genome, whereas the other is the frequency of gene families in the pan-genome. Genes encoding extracellular proteins are at low frequency in species, i.e. each gene family is present in few genomes. This inflates the relative importance of this set of families in pan-genomes. For example, the average genome has ~1% of genes encoding extracellular proteins, but different genomes have distinct families of extracellular proteins. Hence, the frequency of the different gene families of extracellular proteins in the pan-genome is much larger than 1% (around 6 times larger in this case). In contrast, a set of conserved proteins, say the ~50 ribosomal proteins, make 1% of the genome of *E. coli* but a much smaller fraction of the pan-genome (our last pan-genome of *E. coli* had 75 000 proteins, and around 60 ribosomal proteins, making them less than 1‰ of the pan-genome).

Action: We edited the text to make this point clearer.

Fig 1: hard to read black numbers against dark purple background. Purple scaling - % of what?

Answer 1.5: The purple scaling represents the percentage of genomes from a specific clade that encode for extracellular proteins falling within the specific functional category, as it is stated in the figure legend. We agree with the reviewer that the figure was hard to read.

Action: We have changed the colour of the numbers in dark purple background to increase the visibility. We have also modified the legend to ease the readability.

Line 165-167: had a look at the table and there's some dodgy calls for host association at least – eg *H influenzae*, *B pertussis* and *P aeruginosa* are classed as not host associated. What human 16S samples were you using to make these calls? It is understandable that automated analyses of large datasets will produce errors of this kind, but I think worth flagging this class of error in the discussion to illustrate the 'noise' issue.

Answer 1.6: In this study we have used 16S rRNA samples from healthy individuals (to the best of our knowledge), from different tissues, including intestinal, oral, genital and cutaneous. We have also included other hosts in the analysis, such as different

mammals, aves, amphibians, among others. There are here three different cases to consider. 1) The bacterial species is not found. This is typically the case for bacteria like *B. pertussis* that has mostly disappeared from the community because of the existence of a very efficient vaccine. 2) The bacteria is only an opportunistic pathogen and is not usually a commensal of animals. This is the case of *P. aeruginosa* which is not really a pathogen for healthy human individuals and whose frequency as a commensal seems quantitatively negligible (at least in humans). 3) And sometimes there are results that look like errors like for *H. influenzae*, which, even when it is considered as a lung commensal, it is not detected in the selected datasets, thus leaving it unclassified.

Action: We have included a section in the discussion about these issues. We also corrected the supplementary tables to distinguish clearly the values that are 0 (e.g. not soil) from the values that cannot be computed (NA). The statistical analysis accounted for this.

Line 395-399: bit of a non-sequitur here. Authors begin with virulence, but then switch to refs 45-46 which refer to bacteriocins. These kill other bacteria, but are not virulence factors from host perspective.

Answer 1.7: Indeed, this was ambiguous (at best).

Action: We have re-written the paragraph to frame it in the broader topic of antagonistic interactions.

Fig S2: 3 colors not explained clearly

Answer 1.8: We apologize if this was not clearly understood.

Action: We have modified the figure for clarity (and the legend).

Reviewer #2 (Rolf Kümmerli):

Bacteria secrete different classes of compounds to interact with their abiotic and biotic environment. Such compounds include toxins to fight competitors, enzymes to digest food, and secondary metabolites to form biofilms, scavenge metals and swarm on surfaces. There has been tremendous interest in these secreted compounds because they fulfil critical functions for the bacteria themselves, but also for the community as a whole. The latter is the case because secreted compounds can potentially be shared as public goods between cells and thus foster social interactions between them.

While there is an enormous body of experimental work on single-species/single-trait systems in the laboratory, we still know very little on the ecology and evolution of these secreted compounds. In other words, the ecological conditions (abiotic and biotic) that select for compound secretion in natural settings are poorly examined.

The study by Garcia-Garcera and Rocha tackles this issue. The authors conducted a comparative analysis using genomic data (i) to infer the type and abundance of secreted compounds among bacteria and (ii) to quantify community diversity. They further collected data on the habitats the identified species populate. Using phylogeny-controlled analysis, they found that (a) species living in more structured environments secrete more compounds; (b) the diffusion length of secreted compounds is increased in more structured habitats; (c) community diversity correlates with the secretome size, but (as expected) differently for toxins vs. digestive enzymes; (d) there is evidence for

selection for costs optimisation; and (e) generalists secrete more compounds than specialists.

The paper by Garcia-Garcera and Rocha is very well written and easy to follow. The topic is timely, highly relevant, and the insights provided represent a key step towards a better understanding of the evolution and ecology of secreted compounds and community interactions in bacteria. In brief, this is an outstanding paper.

Answer: We thank Dr. Kümmerli for his positive assessment of our manuscript.

Major comments:

(1) The authors should be more careful in explaining the chemistry and biology of secreted compounds. The current notion in the manuscript is “secreted compounds = proteins = public good”. There are several complications, which are currently not addressed in the paper:

a) not everything that is secreted is necessarily a public good. Secreted molecules can be embedded in the membrane of the producer (e.g. biosurfactants) and might only partly be shared with other cells. This fact should be explained and strong statements like ‘extracellular proteins are public goods’ should be revised.

Answer 2.1: We agree that not every extracellular protein is a public good and have now added a caveat concerning that in abstract and in the discussion.

Please note that our text explicitly states that we only analyse extracellular proteins (not every extracellular compound). The pair of words "extracellular protein" was repeated 86 times in our text, it was in the article title, and in several section titles. The word protein was present in 4 of the 5 titles in the section results (it's now present in all of them). So, we think there is no ambiguity in this respect.

The proteins associated with the outer membrane, such as the cases mentioned by the reviewer, were excluded from the analysis. This is now explicitly stated.

b) some secreted compounds can only be shared between cells that have a matching receptor for uptake (e.g. siderophores). Again, this means that such secreted compounds are not necessarily an accessible public good for everyone in the community. This must be explained.

Answer 2.2: " secreted compounds are not necessarily an accessible public good for everyone in the community ". We completely agree and have now mentioned it.

c) not all secreted compounds are proteins. In fact, a high proportion of them are secondary metabolites, where only the machinery for metabolite synthesis is encoded in the genome but not the compound itself. Secondary metabolites include toxins, antibiotics, biosurfactants, siderophores, biofilm matrix components, etc. This important class of secreted compounds is not clearly mentioned in the paper and it is also not clear whether they were included in the analysis. Presumably not. But this must be explained and stated very clearly.

Answer 2.3: We never make the claim that all extracellular products are proteins. This would be obviously wrong, and we actually cite the analyses made for siderophores (which is true, implicitly assumed that the reader knows that siderophores are small molecules, not proteins, which is now made explicit). The entire text is written around extracellular proteins and all the methods are based on the analysis of proteins that are extracellular, so we don't see where there is an ambiguity in our text. We do not analyse secondary metabolites, that's not the expertise of our group, and no such type of analysis is mentioned in the Results or the Methods. We now state explicitly that we do not analyze these compounds in the first paragraph of the introduction. Their study falls outside the scope of this work.

d) with regard to the points above, it must be very clearly stated that the findings of the paper hold for secreted proteins only and not for secreted compounds in general. The dynamics and patterns for secondary metabolites might be quite different. The only other paper that pursued a similar approach is Kummerli et al. 2014 Ecol Lett. (on siderophores). The authors should compare their results to this earlier study and discuss generalities and differences. This would greatly help to obtain a more general understanding of the ecology and evolution of secreted compounds.

Answer 2.4: Again, our text is completely centered around extracellular proteins. But, we do agree that the end of the discussion left some ambiguity regarding the relevance of our results for other extracellular molecules. We have made the text more explicit at the end of the discussion and put forward some thoughts on whether our results would be applicable to small molecules.

(2) In my opinion the general statement that secreted compounds are predominantly encoded on plasmids is wrong. The authors studied proteins, and for this class of secreted compounds the statement might be correct. Within this class, there is a heavy bias towards toxin-antitoxin systems, which are often encoded on plasmids. However, most (if not all) secreted secondary metabolites are encoded on the chromosome. There is a simple reason for this: most secondary metabolites are quorum-sensing controlled and this type of complex regulatory network requires a stable genetic architecture (i.e. the chromosome). This part of bacterial biology is completely neglected in all the papers on 'social traits being enriched on plasmids'. Careful phrasing is required to clarify that the pattern only holds for secreted proteins (and presumably only for toxins).

Answer 2.5: We do not understand the first criticism. We don't make the "general statement that secreted compounds are predominantly encoded on plasmids". First, we only mention extracellular proteins, not secreted compounds. Second, we don't state that extracellular proteins are predominantly encoded on plasmids. What we state is that they are "often encoded in mobile genetic elements (MGEs), most notably in plasmids." And this is clearly indicated by our results. They are strongly over-represented in plasmids, but actually most such proteins are encoded in the chromosome (in integrative mobile genetic elements or not) where they are present at *densities* much smaller than in plasmids. This may have been the basis of this criticism and we now make this point clearer.

About toxins. We disagree with the statement that our results are valid only for protein toxins because plasmids have many toxins-antitoxins (TA). Unfortunately Prof Kummerli does not provide a reference for this statement and we couldn't find any quantitative assessment of this frequency in the literature. To the best of our

knowledge, toxins of TA systems are rarely secreted and many TAs are in chromosomes. Some cases have been reported of toxins of TAs being in the extracellular milieu, because of cell lysis. We only know of one case of active secretion of such toxins (see Lobato-Marquez, FEMS Mic Rev, 2016 for a recent review). Many proteins that are not toxins are secreted into the extracellular space. Actually, figure 1 clearly shows that the proteins we identify have a multitude of different functions.

To definitely clarify this question, we made the following analysis. We queried the proteins classified as extracellular against the database of bacterial exotoxins for humans (DBETH, Chakraborty *et al.* NAR 2012) and the database for Type II toxin-antitoxin systems (Xie *et al.* NAR 2018). Only 2451 (2.1%) of matches could be identified in these databases (protein alignment coverage > 30%). And only 17% of those covered the whole protein (alignment coverage > 90%). Hence, we could find no empirical basis to sustain the claim that extracellular proteins are mostly toxins.

We hope that the text we added to explain the difference between secreted proteins and other compounds will clarify the reviewer and the readers. We opted not to add the analysis of toxins because we believe it will unnecessarily lengthen the text, but will do so if the reviewer and the editor think it's important.

(3) This brings me to my last point. Currently, the authors combine different types of secreted proteins (enzymes and toxins) in most of their analyses. However, I suspect associations to vary between environmental factors and enzymes (providing a cooperative benefit to others) versus toxins (targeting competitors). This is clearly shown in Fig. 4. But what about the other analyses? For instance, it makes sense to evolve diffusible molecules in a structured habitats, as molecules can be shared between nearby relatives. Conversely, toxin production makes little sense in this context as they would only target resistant relatives. There is theory showing that toxins should be more diffusible than beneficial public goods and provide highest benefits at intermediate spatial structure (e.g. Inglis *et al.* 2009 PNAS, Wechsler *et al.* 2019 JEB). I recommend to split all analysis in enzymes vs. toxins as it was done in Figure 4. Introducing a color code for the two classes in the figures might suffice, and report differences in stats if there are any.

Answer 2.6: We have not followed this recommendation for several reasons.

First, as shown in Fig 1, many proteins have unknown function, and most of the proteins of known function are neither bacteriocins nor degradative enzymes. We now make this point clearer. If we restrict all analyses to the categories mentioned in Fig 4 we will ignore the vast majority of the dataset (and strongly reduce the statistical signal and the scope of the study). This is not the point of our study, which aims at studying the repertoires of extracellular proteins in general.

Second, we disagree with the statement that toxins should not select for low diffusivity because they would be useless at close range. There may exist models sustaining this point of view, but there are lots of empirical results showing otherwise. The majority of toxin-based systems that bacteria use to kill other bacteria discovered in the last ten years are used at very close range. It's notably the case of toxins delivered by T6SS, by contact-dependent inhibition systems, and more recently by T4SS (all depend on direct contact between cells).

Third, we tested if the correlation between diffusion length and structure score was valid for each functional category. This, see graph below, is the case for every single functional category.

Fourth, we evaluated the hypothesis that toxins (in this case bacteriocins) are more diffusible at intermediate values. To do so, we have calculated the average diffusion distance per genome for both bacteriocins and degradative enzymes. We observed no statistically significant difference between both groups (Wilcoxon-test p-value = 0.8) (see figure below).

Action: We have added the analysis of the coefficient correlations as supplementary Figure to reassure the reader that the effect is not function-dependent.

Minor comments:

line 35. Start with “We found” to delineate the general intro from the new findings.

Action: We have modified the sentence accordingly

line 39. Unclear what is scavenged.

Action: We have modified the sentence to clarify it.

line 84. Consider rephrasing. Currently it reads as if proteins pick amino acids from a collection for their assembly.

Action: We have modified the sentence to modify its interpretation

line 122 and other places. '16S' should always be spelled out as '16S rRNA'

Action: We have modified the appearance of 16S throughout the whole manuscript.

line 153. Remove hyphens 'carbohydrates' and 'amino acids'

Action: We have modified both words accordingly.

line 163. Statements like this are problematic as they only hold for cooperative proteins, but not for toxins (see major comment 3).

As mentioned above, toxins are a small fraction of the dataset. Also, producers of toxins are producing a public good and are susceptible to cheaters (resistant to toxins) as the producers of other goods.

line 209. reword 'bacteria's habitat'

Action: corrected.

line 308. replace 'trends' by 'effects', as this relationship is significant and not just a trend.

Action: Agreed. We replaced it by "qualitatively similar results" for precision.

lines 346-349. Confusing paragraph. Previously, the authors said that spatial structure favours cooperation, whereas here they state the opposite. I understand the point they want to make, but it might confuse non-specialist readers. Rephrase to something like 'although spatial structure is thought to favor cooperation, it can also have negative effects ...'

Action: We have modified the sentence accordingly.

lines 376-378. I have some doubts (see major comment 2). I believe that this is only true for toxins.

Action: According to our response to major comment 2.5, we have decided to leave this sentence untouched.

lines 401-403. This is one reason why a healthy microbiome can fight off pathogens. This could be mentioned in this context.

Action: We added a sentence and a reference accordingly.

Reviewer #3:

Summary

This paper investigates the effect of community diversity and habitat structure on the repertoire of extracellular proteins in bacteria, with particular focus on extracellular protein frequency and cost. Extracellular protein repertoires are interesting and important as bacteria use them to interact with their environment, including social interactions with other bacteria. The authors use genomic data and 16S environmental datasets to study natural microbial populations.

The main findings are: (i) bacteria living in more structured habitats encode more extracellular proteins; (ii) the effect of community diversity depends on

extracellular protein function: proteins involved with antagonistic interactions (bacteriocins) are more abundant in more diverse communities, whereas proteins involved with nutrient acquisition (degradative enzymes) are more abundant in less diverse communities; (iii) extracellular proteins are less costly in more diverse communities and have lower diffusivity in less structured habitats; (iv) bacterial generalists encode more extracellular proteins than specialists. Overall, the results demonstrate the importance of ecological factors in shaping the evolution of bacterial extracellular gene repertoires.

General comments

This is a strong manuscript with multiple novel and interesting findings. It would be of broad interest to microbiologists and social evolutionary biologists. The authors clearly state their results and the analyses conducted are thorough, using a large genomic sample size. Overall, this paper is a significant advance in our understanding of broad-scale ecological effects on extracellular protein repertoires in bacteria. Some improvements for the manuscript are detailed in the specific comments below.

Answer: We thank the reviewer for these positive assessment of our work.

Specific comments

1. The genome selection process is not explicitly clear from the manuscript (or supplementary material), how was this conducted?

Answer 3.1: In the previous version we included most of the materials and methods in the supplementary annex. We have transferred them back to the main text to facilitate the manuscript readability. We initially included all completely assembled genomes (including plasmids) of the Genbank RefSeq database at the date of November 2016. For all these genomes, we performed the classification of proteins for their sub-cellular location. However, for the sake of the different statistical analyses performed throughout the manuscript, we applied 2 filtering processes: 1. Genomes from phyla including less than 50 genomes were excluded. This includes rare phyla for which usually very few (and most of the times, only one) representatives for each genus are available. Phyla-based analysis lacks power in these cases and the ability of bioinformatics methods to identify extracellular proteins in these phyla is unknown. 2. Genomes whose 16s rRNA classification was ambiguous at the species level, were also discarded from the analysis. If our analysis fails to characterize the link between the genome content and the environmental distribution (through the 16S rRNA information) we could wrongly assign the genomic content of a bacteria to environments where the bacteria was not present. Given the noise associated to the data being analyzed, these two filters were a measure of sanity check to avoid creating false conclusions.

Action: We have rephrased the respective methods section for more clarity.

2. Line 60/61 (and elsewhere) - "Extracellular proteins are public goods that can benefit...". This is not strictly correct, as not all extracellular proteins are public goods. I recommend the authors state that extracellular proteins can act as public goods.

Answer 3.2: This fits the comment 2.1 of reviewer 2. We agree that the statement could be interpreted as a bold generalization.

Action: We have modified the sentence (and elsewhere) accordingly.

3. Lines 214-220 – further alternative hypotheses should be considered when discussing the possible effect of community diversity on extracellular protein frequency. For example, one could hypothesise an increased number of degradative enzymes with increasing community diversity, to increase competitive ability during indirect competition for nutrients.

Answer 3.3: Indeed. But we observe the opposite. Since the reviewer considers we should state this hypothesis, we have now added it to the text of the discussion.

4. Lines 260-261 is lower protein biosynthesis only explained by higher likelihood of exploitation in diverse communities, or could it also be that a lower diversity of secreted protein production is required if each species produces a few that all can then use?

Answer 3.4: This is an interesting hypothesis: bacteria would invest a lot on the production of a few proteins that could be shared across the community. These proteins, because they account for a large fraction of the budget of the cell would be under strong selection to be less costly. However, it could be argued that such bacteria might be very susceptible to exploitation by cheaters and to changes in community composition (if a species is absent this would render some key proteins unavailable). It's worth testing in the lab. But here, testing this hypothesis would require information on the degree of expression of the genes and on the complementary role of extracellular proteins, which we don't have.

Action: none taken.

5. Figure 4. - I understand why the authors have chosen bacteriocins and degradative enzymes to represent antagonistic and cooperative proteins, respectively. However, to further support the authors' data interpretation of Fig 4, results for other public good groups such as nutrient acquisition enzymes would be useful, as degradative enzymes constitute a very broad group of proteins. Did the authors conduct this community diversity analysis on any other classes of extracellular proteins?

Answer 3.5: The list of degradative enzymes that we included in our analysis can be found in the supplementary table 9 and includes enzymes involved in the acquisition of nutrients through the depolymerization of macromolecules, such as cellulose, keratin or starch. It should be noted that in the vast majority of genomes it's not possible at this stage to know if these enzymes are specifically involved in nutrient acquisition.

Action: none taken

6. The discussion is currently quite disjointed. This could be improved by repositioning or reducing the extensive justification for the use of genomic data and the reasons for data dispersion. The discussion also ends abruptly and would be improved if the authors more thoroughly considered the boarder implications of their results.

Answer 3.6: The first reviewer seems to regard the first part of the discussion with the critique to be important and requested its extension. We have done our best to make this part shorter and more complete. We have re-written parts of the discussion to make it more fluid. We also added a last paragraph for openness

7. Fig S3 – is there an explanation for why extracellular proteins from ‘habitat structure 4’ have such a significantly low amino acid cost that contradicts the general trend? Is there anything noticeably different about the genomes in this group?

Answer 3.7: We assume this comment relates to the strange boxplot above #4 in Fig S4. This intrigued us for a while. Now, after more thorough analysis, we believe we have understood the reason. These bacteria are soil-associated, which has been shown to be associated with high G+C content genomes (Foerstner *et. al.* 2005 EMBO rep). It is well-known that the universal genetic code leads to an enrichment of G+C rich genomes in less expensive amino acids (Seligmann 2003 J Mol. Evol.). When we analysed our genome data, we could confirm these previous results. Bacteria across the different habitats have similar average G+C contents (note that G+C contents vary a lot for each habitat, but the averages are not significantly different). The only exception is soil, where the genomic G+C is higher and cost is lower both in extracellular and in non-extracellular proteins (hence, the effect is not associated with the object of this study). Overall, this effect is not sufficient to affect the global correlation between cost and habitat structure.

Action: We have added this analysis to the results section.

Reviewers' Comments:

Reviewer #1:

Remarks to the Author:

The authors have carefully responded to all of my earlier critiques

Reviewer #2:

Remarks to the Author:

I have now carefully studied the revised manuscript by Garcia-Garcera and Rocha and their responses to my comments. I was already very positive during the first round of review, and I'm still positive at this stage: the paper is great work. The authors have addressed some of my comments and I'm happy with the implemented changes.

However, I was also surprised by the harsh tone of certain responses and how reluctant the authors were to constructively discuss my inputs. Maybe this happened because I did not clearly explain my thoughts. So I reiterate:

1. I refer to answer 2.5 and the authors' statement "The genes encoding these proteins have some peculiarities. First, they tend to be gained and lost at a high rate, relative to other genes, and are often encoded in mobile genetic elements (MGEs), most notably in plasmids" (lines 68-70). I understand this statement and I know the underlying comparative papers very well. My point is that I believe that this statement is primarily true for secreted toxins, and might be less valid (or the association less strong) for non-toxin extracellular proteins. The authors replied: "Unfortunately Prof Kummerli does not provide a reference for this statement." - This is correct. There exists no reference. The point is that the authors have now the unique opportunity to test this hypothesis. They could split their data set as they did in Figure 4 and ask whether both classes of proteins are overrepresented on MGEs. I predict they are not. My reasoning is that the expression of degradative enzymes is usually controlled by a complex regulatory network (e.g. quorum sensing), which cannot easily be transferred between strains/species on MGEs.

2. I refer to answer 2.6. Here, I made the suggestion that certain environmental effects could be different for degradative enzymes as opposed to toxins. I particularly referred to theoretical work showing that secreted toxins are most beneficial at intermediate strain mixing (i.e. habitat structure score) (Inglis et al. 2009). The authors would now have the unique opportunity to test this theory. For instance, they could repeat their analysis for toxins and degradative enzymes as they did in Fig. 2 and check whether the relationship become humped-shaped as opposed to linear for toxins.

Instead, the authors discounted my idea: "Second, we disagree with the statement that toxins should not select for low diffusivity because they would be useless at close range. There may exist models sustaining this point of view, but there are lots of empirical results showing otherwise. The majority of toxin-based systems that bacteria use to kill other bacteria discovered in the last ten years are used at very close range. It's notably the case of toxins delivered by T6SS, by contact-dependent inhibition systems, and more recently by T4SS (all depend on direct contact between cells)." - This response confused me because their study is not about contact-dependent delivery systems, but about secreted diffusible toxins. And for this group of toxins, the theoretical predictions are clear.

3. Finally, I have one point that refers to response 1.1, to a comment made by reviewer 1. This is a valid point and classification of habitat structure is not easy. In our Kummerli et al. 2014 Ecol Lett paper, we performed robustness analyses, including the collapse and shuffling of adjacent categories.

The authors did something similar here and I believe their approach is robust.

Minor comments:

line 341: I'm not sure I understand "To tackle this, we used quality controls, resampling and literature (to identify specialist)." What were these quality controls, and which literature was used? Please specify.

line 352: "was lacking in all strains" is unclear. Do the authors mean that B. pertussis was not found in the data sets? Please clarify.

Reviewer #3:

Remarks to the Author:

I have no further comments on the manuscript and am satisfied with the responses to the reviewers and the relevant changes made.

REVIEWERS' COMMENTS (second round):

Reviewer #1:

The authors have carefully responded to all of my earlier critiques

Answer 1.1: We would like to thank once more the reviewer for the positive assessment of the manuscript and the improvements made thanks to his/her previous comments.

Reviewer #2:

I have now carefully studied the revised manuscript by Garcia-Garcera and Rocha and their responses to my comments. I was already very positive during the first round of review, and I'm still positive at this stage: the paper is great work. The authors have addressed some of my comments and I'm happy with the implemented changes.

Answer 2.1: We would like to thank again Prof. Kümmerli for his positive and thorough review.

However, I was also surprised by the harsh tone of certain responses and how reluctant the authors were to constructively discuss my inputs. Maybe this happened because I did not clearly explain my thoughts.

Answer 2.2: We would like to apologize if our tone was taken as harsh, as it was not our intention. We regarded some statements of the previous review as not clear enough (or lacking literature support), while tackling them took a lot of time and effort (even though many controls had already been done). We also have the impression that the reviewer is spurring us to test other theories and other ideas that are peripheral to the main message of the study. We agree that these are interesting, but this paper is already quite long and we don't want to blur its message by focusing on other theories or small subsets of the data. Many of these interesting ideas can be tested in the future when more data (and a grant) is available for it. We think the manuscript already provides a lot of novel information and we believe that should be enough to grant its publication.

Comment 1. I refer to answer 2.5 and the authors' statement "The genes encoding these proteins have some peculiarities. First, they tend to be gained and lost at a high rate, relative to other genes, and are often encoded in mobile genetic elements (MGEs), most notably in plasmids" (lines 68-70). I understand this statement and I know the underlying comparative papers very well. My point is that I believe that this statement is primarily true for secreted toxins, and might be less valid (or the association less strong) for non-toxin extracellular proteins. The authors replied: "Unfortunately Prof Kummerli does not provide a reference for this statement." - This is correct. There exists no reference. The point is that the authors have now the unique opportunity to test this hypothesis. They could split their data set as they did in Figure 4 and ask whether both classes of proteins are overrepresented on MGEs. I predict they are not. My reasoning is that the expression of degradative enzymes is usually controlled by

a complex regulatory network (e.g. quorum sensing), which cannot easily be transferred between strains/species on MGEs.

Answer 2.3: As we stated in our previous answers, the main limitation we encounter to Prof. Kümmerli's proposal is the limited number of genomes with enough extracellular proteins associated to each category in both chromosome and plasmids, as both categories are quite specific. This impairs the possibility to compare both functional categories with enough statistical power to extract significant conclusions out of it. Nevertheless, to assess the claim that only toxins are more associated to plasmids, which is secondary to the main argument of the manuscript, we have compared the frequency of each functional category represented in figure 1 in chromosomes and plasmids. Except for "Coenzyme metabolism" and "Translation" (for which the statistical power is very low due to lack of candidates in plasmids), we found a significant over-representation of secreted proteins in plasmids across every functional category (Wilcoxon rank test $P < 0.0006$, after FDR). These results support our claim that extracellular proteins (irrespective of their function) are encoded at higher frequency in plasmids.

We would like to emphasize at this stage that the goal of this paper is not to show that secreted proteins are often encoded in mobile elements. We have shown this to be generically true before in several publications. It is also not aimed to focus on two specific functions that account (to the best of our knowledge) for a small fraction of all secreted proteins. We therefore only added a sentence to our text to mention this new result. We hope this convinces the reviewer that a lot of different functions of secreted proteins are over-represented in plasmids.

2. I refer to answer 2.6. Here, I made the suggestion that certain environmental effects could be different for degradative enzymes as opposed to toxins. I particularly referred to theoretical work showing that secreted toxins are most beneficial at intermediate strain mixing (i.e. habitat structure score) (Inglis et al. 2009). The authors would now have the unique opportunity to test this theory. For instance, they could repeat their analysis for toxins and degradative enzymes as they did in Fig. 2 and check whether the relationship become humped-shaped as opposed to linear for toxins.

Answer 2.4: As explained in the previous answer (2.3), the analysis proposed cannot be performed in the current dataset due to sample size limitations. In this case, the limitation is much greater as we would need to restrict our analyses to the species found in single environments. We envisage that with the increasing rate of genome sequencing, and a better functional characterization and environmental classification, we might be able to answer prof. Kümmerli's proposed hypothesis. However, the available evidence suggests that the hypothesis is false, since the category of secreted proteins involved in carbohydrates metabolism is over-represented in plasmids relative to the chromosome (information indicated in the second paragraph of the results in the old and in the new versions of the text).

Instead, the authors discounted my idea: "Second, we disagree with the

statement that toxins should not select for low diffusivity because they would be useless at close range. There may exist models sustaining this point of view, but there are lots of empirical results showing otherwise. The majority of toxin-based systems that bacteria use to kill other bacteria discovered in the last ten years are used at very close range. It's notably the case of toxins delivered by T6SS, by contact-dependent inhibition systems, and more recently by T4SS (all depend on direct contact between cells)." - This response confused me because their study is not about contact-dependent delivery systems, but about secreted diffusible toxins. And for this group of toxins, the theoretical predictions are clear.

Answer 2.5: First, we discarded the proposed analysis due to the reasons explained in answers 2.3 and 2.4 (insufficiently detailed data and the one that is available suggests the hypothesis is false).

Second, Prof. Kümmerli states in his first review that "toxin production makes little sense in this context (structured environments) as they would only target resistant relatives". Overall, Prof. Kümmerli's review included references to concepts with a much broader scope than those included in our manuscript (i.e. Secondary metabolites, toxin-antitoxins, etc...). For such reason, our response also included a broader level of discussion, such is the case of introducing contact-dependent toxins (which we agree, it is not the focus of our study) to exemplify that the evolution of warfare has many examples of close-range antagonism.

3. Finally, I have one point that refers to response 1.1, to a comment made by reviewer 1. This is a valid point and classification of habitat structure is not easy. In our Kummerli et al. 2014 Ecol Lett paper, we performed robustness analyses, including the collapse and shuffling of adjacent categories. The authors did something similar here and I believe their approach is robust.

Answer 2.6: We thank prof. Kümmerli for his support on this aspect of the manuscript, which is probably the most difficult to control.

Minor comments:

line 341: I'm not sure I understand "To tackle this, we used quality controls, resampling and literature (to identify specialist)." What were these quality controls, and which literature was used? Please specify.

Answer 2.7: The different quality controls, resampling processes and literature revision (including literature references) are widely explained throughout the methods section. In brief, we have controlled all our analysis by different genomic features (such is the GC content, the genome/chromosome sizes, etc...). Second, we have applied resampling when necessary, which includes cases where different categories included a significant number of elements (i.e. different number of species associated to different environmental structures). Third, we have included Bergey's as reference for the specialists, due to the limitation on the number of references allowed in the manuscript.

line 352: "was lacking in all strains" is unclear. Do the authors mean that *B. pertussis* was not found in the data sets? Please clarify.

Answer 2.8: Yes, as explained in the answer 1.6. of the previous review, *B. pertussis* was not found in any of the datasets. Same occurs for other bacterial

species for which we had the genome, but we couldn't find in the 16S rRNA datasets. We changed the text to make it clearer.

Reviewer #3:

I have no further comments on the manuscript and am satisfied with the responses to the reviewers and the relevant changes made.

Answer 3.1: We would like to thank the reviewer for the careful review of the manuscript.